# Topographic Complexity Is a Principal Driver of Plant Endemism in Mediterranean Islands

**DOI:** 10.3390/plants13040546

**Published:** 2024-02-17

**Authors:** Leanne Camilleri, Katya Debono, Francesca Grech, Andrea Francesca Bellia, Gyllain Pace, Sandro Lanfranco

**Affiliations:** Department of Biology, University of Malta, MSD2080 Msida, Malta; leanne.camilleri@um.edu.mt (L.C.); katya.debono.22@um.edu.mt (K.D.); francesca.grech.19@um.edu.mt (F.G.); andrea-francesca.bellia@um.edu.mt (A.F.B.); gyllain.pace.23@um.edu.mt (G.P.)

**Keywords:** topographic complexity, Mediterranean islands, species-area relationships

## Abstract

The frequency of endemism in the flora of Mediterranean Islands does not follow a straightforward species–area relationship, and the determinants of endemism are several and complex. The aim of this study was to estimate the explanatory power of a small number of variables on the species richness of vascular plants on selected Mediterranean islands and archipelagos, and on the proportion of narrow endemism in each. We used a novel approach whereby the topographic complexity and isolation of an island were estimated through more detailed methods than those utilised previously. These parameters, along with island area and human population density, were used in a number of regression models with the number of taxa or proportion of endemics as the dependent variables. The results demonstrated that ‘topography’, a factor that was not specifically included in previous models for Mediterranean islands, exerted a consistent, statistically significant effect on both the number of taxa as well as the proportion of endemic taxa, in all models tested. The ‘isolation’ factor was not a significant predictor of the number of taxa in any of the models but was a statistically significant predictor of the proportion of endemic taxa in two of the models. The results can be used to make broad predictions about the expected number of taxa and endemics on an island, enabling the categorisation of islands as ‘species-poor’ or ‘species-rich’, potentially aiding conservation efforts.

## 1. Introduction

The Mediterranean region harbours an estimated 25,000 species of vascular plants [1,2,3], representing approximately 7% of known plant species in only 1.6% of the global land area [4]. Approximately 60% of these species are endemic to the Mediterranean region [1,2], with more than half of these regional endemics (c. 37% of all Mediterranean species) being narrow endemisms restricted to a single well-delineated area within the region [3]. This disproportionately high alpha, beta, and gamma diversity makes the Mediterranean region one of 36 ‘biodiversity hotspots’ identified as priority areas for conservation [5,6]. There are multiple reasons for this high plant diversity, including the geological and climatic history, heterogeneity of substrata, presence of mountainous areas, and numerous islands. Many of these contexts favour speciation processes, generating the anomalously high diversity observed.

Within the Mediterranean hotspot, islands are a conspicuous contributor to plant diversity. The geological history of the region has formed approximately 11,000 islands [7], many of them continental. The five largest islands (Sicily, Sardinia, Corsica, Crete, Cyprus) and one archipelago (Balearics) were included in a list of Mediterranean biodiversity hotspots [8], suggesting that an understanding of the processes influencing the accumulation of species on Mediterranean islands and the evolution of some of these into endemic taxa, is of prime importance for informed conservation strategies in the region.

The relationship between the geospatial properties of an island and its species richness has attracted much attention over several decades. The basic tenets of the theory of island biogeography [9] relate the number of species (S) of an island to its area (A) in a non-linear manner described by: S = cA^z^, with c and z being constants specific to the island system being studied. This simple expression provides an initial basis for the prediction of equilibrium species numbers but does not make any predictions about endemism on islands. In general, the formation of endemisms would depend on geographical contexts that favour reproductive isolation of subpopulations from a parent population. As such, the basic geospatial properties of an island can be reviewed in the light of their tendency to reduce gene flow between isolated subpopulations. The area of an island, whilst theoretically correlated with the number of taxa, is, a priori, not in itself expected to be a main driver of speciation. The proximity of an island to a source of propagules (a biogeographical ‘mainland’) is expected to be a relevant factor in that an isolated island is less likely to receive regular gene flow from the mainland. On an island itself, a strong elevational gradient would create different climatic conditions within a relatively small area, favouring parapatric speciation. It would also be expected that topographic complexity would also be a very important factor in isolating subpopulations and reducing gene flow between them.

The classic Arrhenius species-area model has been modified [10] to incorporate a term that unifies the area per se and the habitat diversity. This model, named the ‘choros’ model by its authors, gave a better mathematical fit in most of the cases tested [10] and has since been used in other studies. The present study takes a comparable approach in that it introduces a term reflecting habitat heterogeneity, although it has been given a different label.

Many of these conditions have been considered for Mediterranean islands and archipelagos. Various works [11,12] have studied, or alluded to, the positive relationship between island area and species richness in the Mediterranean. Moreover, [13,14,15,16,17] also evaluated the effect of the maximum elevation of an island with species richness. As regards ‘isolation’, [18] recorded no significant correlation between species number and the distance to the mainland in Aegean islands. In general, however, the tendency of such studies is to simplify complex biogeographic drivers into unitary, easily measurable parameters, such as ‘distance’ or ‘elevation’, potentially transferring some of the explanatory power of their models into the ‘unexplained variation’ component. This represents a knowledge gap that this study will address.

It should be borne in mind that the geospatial properties of an island or archipelago are not the only drivers of speciation, as ecological interactions within islands can modulate the predicted numbers of species. However, over the past 10,000 years, these interactions have mainly been dwarfed by the impact of humans on the Mediterranean environment [4,7]. The archaeological record suggests that humans have simplified and largely homogenised natural habitats through their conversion into agricultural landscapes, creating ‘new’ urban and suburban environments that exclude many indigenous species and favour the introduction of xenophytes and the formation of hybrids, therefore reducing ‘effective isolation’ through a higher frequency of long-distance travel and the deliberate and passive transport of propagules across the region. As such, human population size on an island is likely to be an important factor regulating opportunities for speciation. The use of the term ‘human impact’ is likely to be misleading, as it implies a unitary perception of a large and complex network of processes, making its representation in models an attractive proposition. However, the precise articulation of this vague set of processes into a mathematical model is far from straightforward and any proxies used (such as ‘population density’) would be considered oversimplifications.

The aim of this study is to estimate the explanatory power of a small number of compound variables on the species richness of vascular plants on selected Mediterranean islands and archipelagos, and on the proportion of narrow endemism in each. Although this, in itself, is not new, we propose a novel approach whereby the topographic heterogeneity and isolation of an island are estimated through more detailed methods than those utilised up to this point. Although the study aims to derive a mathematical relationship between the proportion of endemics on an island and other environmental factors, it is understood that this relationship cannot be interpreted as a completely deterministic one. As such, the trends derived from the data will be used to highlight the relative importance of various parameters or processes on endemism in Mediterranean islands.

## 2. Results

### 2.1. Basic Trends

Exploratory analysis of the data suggested that the total number of taxa on each island increased with the surface area (r = 0.834, *p* < 0.05) and with increasing topographic complexity (r = 0.902, *p* < 0.01). In accordance with the basic equation of island biogeography, the preliminary analysis was repeated after the surface area and number of taxa were log-transformed. This transformation increased the value of the correlation coefficient and level of statistical significance (r = 0.850, *p* < 0.01). Given the marginal increase in the value of r, the original untransformed data was utilised in subsequent analyses. The number of endemic taxa on each island was also correlated with increasing topographic complexity (r = 0826, *p* < 0.05) although this trend should be interpreted with caution, as the number of endemic taxa is fundamentally dependent on the number of taxa and no causative interpretation can be drawn at this stage.

### 2.2. Predictive Models on the Number of Endemic Taxa

The effect of the four predictors on the number of taxa on each island or archipelago was examined through the construction of five negative binomial regression models, labelled A1 to E1. The characteristics of these models are summarised in Table 1. In all five models, ‘topography’ and ‘area’ exerted a significant effect on the number of taxa. Neither the ‘population’ nor ‘isolation’ of the island or archipelago exerted effects that were significantly different from a random effect in all the models they were included in. The model with the lowest AICc was C1, relating the number of taxa to two predictors (Topography and Area). This model also had the lowest AICc and explained >96% of the variability in the number of taxa. A generalised linear model (GLM) plot of this model is shown in Figure 1.

### 2.3. Predictive Models on the Proportion of Endemic Taxa

The effect of the four predictors on the proportion of endemic taxa on each island or archipelago was examined through the construction of four beta regression models, labelled A2 to D2. The characteristics of these models are summarised in Table 2.

The full model (A2) returned the highest Pseudo R^2^ but also the highest AICc. In this model, all explanatory variables were significant predictors of the proportion of endemic taxa, with the effects of ‘isolation’ and ‘area’ being larger than those of ‘population’ and ‘topography’. Model B2, with three geospatial explanatory variables, returned a much lower AICc with a lower Pseudo R^2^. ‘Topography’ and ‘isolation’ were significant predictors of the proportion of endemic species, with the effects of ‘topography’ exerting the greater effect. The effect of ‘area’ was not significantly different from a random effect. The simpler models (C2 and D2) had a lower AICc but with much lower Pseudo R^2^ values, with only ‘topography’ returning a significant effect.

## 3. Discussion

Geospatial drivers: The area of an island or archipelago was a significant determinant of the number of taxa in all the models where it was included; conversely, it was only a significant predictor of the proportion of endemic taxa in the full model (A2) but not in the geospatial models B2 and C2. The results clearly demonstrated that ‘topography’, a factor that was not specifically included in previous models for Mediterranean islands, except as ‘choros’ [10], exerted a consistent, statistically significant effect on both the number of taxa as well as the proportion of endemic taxa, in all models tested. As with [18], the ‘isolation’ factor was not a significant predictor of the number of taxa in any of the models but was a statistically significant predictor of the proportion of endemic taxa in two of the models (A2 and B2).

In broad terms, these results suggest that immigration rates onto an island depend on the size of the receiving habitat and the diversity of potential niches it supports. In this case, ‘isolation’ is presumably not a significant determinant of the number of taxa as, given sufficient time, propagules from more distant sources would still occasionally reach the island through infrequent dispersal events (unusual anemochoric events such as freak storms, unusual ornithochoric events such as stray migrants etc.) [19,20].

Conversely, the drivers of speciation (and therefore, of endemism) are dependent on the interruption of gene flow between populations. This is favoured by the presence of ecological ‘islands’, such as mountain peaks, deep ravines, and other isolated geomorphological contexts, all of which contribute to the topographic complexity. ‘Isolation’ would now be an important factor, since the reduced immigration rate characteristic of isolated islands would be reflected in reduced gene flow from the mainland, also reducing the probability of genetic dilution of island variants. A varied topography can also favour the persistence of relict populations in refugia [21], promoting the formation of ‘restricted endemics’ and ‘disjunct endemics’ [3,12,22,23].

Human population density: Human population density did not have a significant effect on the number of taxa and only influenced the proportion of endemic taxa significantly in the full model (A2). In biogeographical terms, this is predictable, as human activity is unlikely to prevent the initial colonisation of an island. However, the impact of human activity may lead to the simplification of existing habitats, reducing the niches available for speciation, as well as the possible extirpation of small populations. Moreover, human activity is also correlated with the presence of xenophytes [24,25], some of which may be invasive alien species, further compressing the fundamental niches of endemic species through interspecific competition. Considering ‘human activity’ as a unitary predictive factor is problematic, as it amalgamates multiple independent processes into a single factor. The only common denominator is the cause, rather than the effect itself, and may give a misleading impression of homogeneity that is concealing a broad heterogeneity of effects.

## 4. Materials and Methods

### 4.1. General Structure of the Study

The study consisted of the compilation of basic geospatial, taxonomic, and human population parameters for a sample of islands and archipelagos in the western and central Mediterranean regions. These data were then used to model the relative contribution of each parameter towards the variance in the number of taxa and proportion of endemic taxa. The analysis focused on ‘taxa’ rather than ‘species, as it reduces the subjective recognition of taxonomic ranks between species and subspecies.

### 4.2. Choice of Islands and Archipelagos

This study focused on a small subset of the over 11,000 islands [7] in the Mediterranean region. The sample consisted of the five largest islands (Sicily, Sardinia, Corsica, Crete, and Cyprus) and was augmented by the western-central Mediterranean archipelagos including the Balearic Islands, the Tuscan archipelago and Malta, some of which share a common origin [26]. The choice of islands and archipelagos was designed to span a spectrum of island sizes and types. Archipelagos were not merely identified on the mutual proximity of the constituent islands but also on their shared origins and geological histories. All these factors combine to make the constituent islands of an archipelago floristically more similar to each other than islands in other archipelagos. Naturally, this does not exclude the overlap (sometimes considerable) between islands in different island groups.

### 4.3. Geospatial Properties

Surface area: The surface area of each island or archipelago in the sample was obtained from the Global Self-consistent, Hierarchical, High-resolution Geography Database (GSHHG) (https://www.soest.hawaii.edu/pwessel/gshhg/ accessed on 4 December 2023).

Topographic complexity: The present study extended the notion of ‘maximum elevation’ used in other studies [13,18,27,28] into that of ‘topographic complexity’ as it captures the drivers behind elevational habitat diversity more accurately than a single point measurement would. The topographic complexity of each island was estimated using an “Index of Roughness” [29]. This was calculated by first downloading a base map of an entire island from Google Earth Pro (Google, n.d.) and subsequently superimposing a standard grid over the map using ImageJ v.1.54h [30]. The topographic elevation, in metres above mean sea level, was read from the Google Earth Pro digital elevation model at each intersection point of the grid. Topographic complexity was defined as the arithmetic mean of the absolute differences between each data point and its immediate neighbours, reflecting changes in elevation between adjacent points. For edge data points, only the available neighbours were used. Although this method provides a scale-dependent estimate, it was considered an improvement over approaches that simplified elevational diversity as either the maximum elevation of an island or as the range of elevations within a given area.

Isolation: The ‘isolation’ of each island from neighbouring land areas, presumed to be possible sources of incoming propagules, was calculated by preparing an image for each island, centred on the island of interest, and showing all the land within a radius of 300 km from the centre of the island. The images were subsequently processed and calibrated in ImageJ v.1.54h and used to integrate the total area of neighbouring land within this radius for each island. The 300 km radius was selected after an iterative process during which the radius was increased in increments of 100 km. It was the smallest scale at which any of the islands or archipelagos in the sample were completely circumscribed. This method is similar in concept to the ‘surrounding landmass model’ [31]. This procedure gives a metric of ‘proximity’ of the island concerned to surrounding landmasses. It was converted to its conceptual inverse, ‘isolation’, by calculating the area not occupied by the surrounding landmasses in the zone of survey.

### 4.4. Human Population Data

Recent population sizes for each island were obtained from national statistical agencies: ISTAT for Sardinia, Sicily, and the Tuscan Archipelago, ELSTAT for Crete, Institut National de la Statistique et des Études Économiques for Corsica, caixabankresearch.com for the Balearics, eurydice.eacea.ec.europa.eu for Cyprus, and the National Statistics Office for Malta. Population density was subsequently derived through a straightforward calculation. For the purposes of this study, it was assumed that the population density had a comparable direct linear relationship with ‘human impact’ over all islands.

### 4.5. Taxa

The total number of taxa and the number of restricted endemic taxa (species with a range limited to just one small, well-defined area of extent [3]) for each island were obtained from the most recent published sources specific for that island. These included [32] for Corsica, [33] for Crete, [28] for Sicily, [34] for Sardinia, [35] for Cyprus, [17,36] for the Balearics, [27,37,38] for the Tuscan Archipelago, and [39] for Malta. Not all sources were equally recent, and this should be considered as a possible source of error. However, given that changes in the number of total species and endemic species are unlikely to be large compared to the recorded number of taxa, this is not expected to influence the statistical significance of the results.

### 4.6. Notation Used in Models

The parameters described above were abbreviated as follows in models:

‘Area’: surface area of the island or archipelago, in km^2^.

‘Topography’: Topographic complexity, in m.

‘Isolation’: The cumulative area of land withing 300 km of the island or archipelago, in km^2^.

‘Population’: human population density, in individuals km^−2^.

‘Spp’: total number of taxa (count data).

‘Prop’: proportion of endemic taxa (compositional data).

The data compiled for each island or archipelago, and on which the analyses were based, are summarised in Table 3.

### 4.7. Management and Analysis of Data

All analyses were carried out using R [40] implemented in RStudio, version 2023.12.0.369 and in CANOCO, version 5.15 [41]. The relationship between the number of taxa on an island or archipelago and the predictor variables (area, isolation, topographic complexity, human population density) was tested using a series of negative binomial regression models, a method appropriate for count data where the variance is much larger than the mean, as was the case for the data in this study [42]. The relationship between the same predictor variables and the proportion of endemic taxa was explored using beta regression, a method appropriate for compositional data [43,44]. Prior to analysis, conformity to the assumptions of the methods was assessed visually using the Ggally R package [45] whilst residuals diagnostics were examined using the DHARMa R package [46]. In cases with a number of taxa, the models that explained the most variance were selected through best-subsets regression and their relative performance compared using the Akaike Information Criterion corrected for small sample sizes (AICc).

## 5. Conclusions

The results of this study have recapitulated the importance of island area in determining the number of taxa but have also highlighted the fundamental significance of topographic complexity as a factor determining both the number of taxa as well as the proportion of endemic species. Geographic isolation was found to be relevant in predicting the proportion of endemic species but not for the number of taxa on an island. The results can be used to make broad predictions about the expected number of taxa and endemics on an island, enabling the categorisation of islands as ‘species-poor’ or ‘species-rich’ [47], potentially aiding conservation efforts.

## Figures and Tables

**Figure 1 plants-13-00546-f001:**
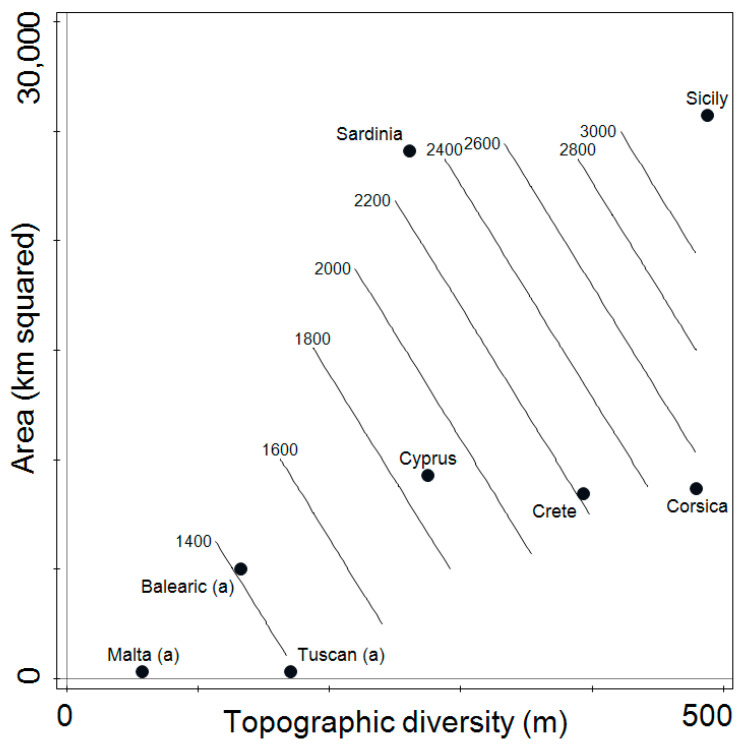
GLM plot of model C1 (count of endemic taxa as a function of the predictors “surface area of island” and “topographic diversity of island”); explained deviance = 96.3%). The contour lines represent an estimate of the total number of taxa in each zone of the plot. The term (a) indicates that the site being considered is an archipelago. https://github.com/SandroLanfranco/endemics.git (accessed on 12 February 2024).

**Table 1 plants-13-00546-t001:** Properties of negative binomial regression models A1, B1, C1, D1, and E1. Predictor variables with statistically significant effects are shown in bold font.

Model	Predictors, p(T)	Null Deviance	Residual Variance	AICc
A1	Area, <0.001 **Topography, <0.0001** Isolation, 0.416 Population, 0.142	272.30	7.96 (97.1%)	110.24
B1	**Area, <0.0001** **Topography, <0.0001** Isolation, 0.753	216.50	8.08 (96.3%)	110.18
C1	**Area, <0.0001** **Topography, <0.001**	216.07	8.07 (96.3%)	108.2
D1	**Topography, <0.0001**	48.32	8.01 (83.4%)	118.1
E1	**Area, <0.0001**	23.70	8.05 (65.3%)	123.85

**Table 2 plants-13-00546-t002:** Properties of beta regression models A2, B2, C2, and D2. Predictor variables with statistically significant effects are shown in bold font.

Model	Predictors, p(T)	Pseudo-R^2^	AICc
A2	**Area, <0.001** **Topography, 0.002** **Isolation, <0.001** **Population, 0.003**	0.851	53.2
B2	Area, 0.09 **Topography, <0.001** **Isolation, 0.015**	0.684	4.8
C2	Area, 0.897 **Topography, 0.05**	0.431	−8.9
D2	**Topography, 0.017**	0.423	−18.2

**Table 3 plants-13-00546-t003:** Geospatial, population, and taxonomic data for each island. Archipelagos are indicated by the suffix “(a)”. Explanation of the column headers is in the text.

Island	Topography	Isolation	Area	Population	Spp	Endemics	Prop
Corsica	480	0.691	8682	30	2680	284	10.6%
Crete	394	0.952	8450	75	2214	392	17.7%
Sardinia	261	0.963	24090	65	2438	195	8.0%
Sicily	488	0.946	25711	186	3252	293	9.0%
Cyprus	275	0.736	9251	120	1649	146	8.9%
Malta (a)	58	0.896	316	1715	1100	26	2.4%
Balearic (a)	133	0.880	4992	237	1551	145	9.3%
Tuscan (a)	171	0.522	313	110	1400	17	1.2%

## Data Availability

Raw and processed data, analytical workflows, and R scripts will be made available from the corresponding author on request. They are also available at https://github.com/SandroLanfranco/endemics.git.

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
