# Peer review of "Topographic Complexity Is a Principal Driver of Plant Endemism in Mediterranean Islands"

_plants, 2024, doi:10.3390/plants13040546_

Round 1
Reviewer 1 Report
Comments and Suggestions for Authors
The manuscript entitled “Topographic complexity is a principal driver of plant endemism in Mediterranean Islands” is an interesting research that presents alternative approaches to investigate the main drivers of endemic, and not only, species richness. The main findings are that topography, and especially how it was calculated, is the main driver in determining endemic richness.
The manuscript is well and concisely written, scientifically sound and potentially of interest to the readers of Plants Journal. Given its interest, I recommend expanding the methodological description and discussion. To this end, I suggest several recent and older references to support both the introduction and discussion.
Please, see specific comments for details.
Abstract:
General comment: I recommend mentioning at the end of the abstract the main conclusions
L 21 a missing space after the dot
L 21 ‘clearly’ can be deleted
Introduction
L 36: currently, up to 36 hotspots are commonly recognised
Hrdina A, Romportl D (2017) Evaluating global biodiversity hotspots–very rich and even more endangered. J Landsc Ecol 10:108–115
https://www.cepf.net/our-work/biodiversity-hotspots/hotspots-revisited
L 39: I’d delete the term ‘allopatric’ as it is not always true
L 42: delete ‘itself’?
L 64: please include references to support this sentence, many of those cited below can be used for this one claim
L 71: most of the cited papers (maybe apart from 11) are simple floras of a specific island or nor even analysing any aspect related to species richness (especially 8), I suggest considering the following papers:
Chiarucci, A., Guarino, R., Pasta, S., Rosa, A. L., Cascio, P. L., Médail, F., ... & Zannini, P. (2021). Species–area relationship and small‐island effect of vascular plant diversity in a young volcanic archipelago. Journal of Biogeography, 48(11), 2919-2931.
Fois, M., Fenu, G., & Bacchetta, G. (2016). Global analyses underrate part of the story: finding applicable results for the conservation planning of small Sardinian islets’ flora. Biodiversity and Conservation, 25, 1091-1106.
Kougioumoutzis, K., Kokkoris, I. P., Panitsa, M., Kallimanis, A., Strid, A., & Dimopoulos, P. (2021). Plant endemism centres and biodiversity hotspots in Greece. Biology, 10(2), 72.
Testolin, R., Attorre, F., Bruzzaniti, V., Guarino, R., Jiménez‐Alfaro, B., Lussu, M., ... & Chiarucci, A. (2023). Plant species richness hotspots and related drivers across spatial scales in small Mediterranean islands. Journal of Systematics and Evolution.
L 72 similarly, I suggest considering the following papers that related endemic richness and elevation, some of them are already cited in other parts of the text
Guardiola, M., & Sáez, L. (2023). Are Mediterranean Island Mountains Hotspots of Taxonomic and Phylogenetic Biodiversity? The Case of the Endemic Flora of the Balearic Islands. Plants, 12(14), 2640.
Kazakis, G., Ghosn, D., Remoundou, I., Nyktas, P., Talias, M. A., & Vogiatzakis, I. N. (2021). Altitudinal vascular plant richness and climate change in the alpine zone of the Lefka Ori, Crete. Diversity, 13(1), 22.
Fois, M., Fenu, G., Cañadas, E. M., & Bacchetta, G. (2017). Disentangling the influence of environmental and anthropogenic factors on the distribution of endemic vascular plants in Sardinia. PloS one, 12(8), e0182539.
Steinbauer, M. J., Irl, S. D., & Beierkuhnlein, C. (2013). Elevation-driven ecological isolation promotes diversification on Mediterranean islands. Acta Oecologica, 47, 52-56.
L 74: in Fois et al. (2016), endemic species richness was inverse to isolation, measured as 'surrounding land mass around the island perimeter' (SLMP). This index, proposed in Weigelt & Kreft (2013, see SI Materials and Methods) is similar to the one used here and applied elsewhere (e.g. Fois et al. 2016; Sandel et al. 2019). Thus, it might be mentioned also when describing their metric of isolation in Methods and compared to other results in the discussion.
Weigelt P, Kreft H (2013) Quantifying island isolation–insights from global patterns of insular plant species richness. Ecography 36:417–429.
Fois, M., Fenu, G., & Bacchetta, G. (2016). Global analyses underrate part of the story: finding applicable results for the conservation planning of small Sardinian islets’ flora. Biodiversity and Conservation, 25, 1091-1106.
Sandel, B., Weigelt, P., Kreft, H., Keppel, G., van der Sande, M. T., Levin, S., ... & Knight, T. M. (2020). Current climate, isolation and history drive global patterns of tree phylogenetic endemism. Global Ecology and Biogeography, 29(1), 4-15.
L 151: for clarity, I'd change the title of this subchapter into something like "predictive models on the proportion of endemic taxa". The same for the previous one
Discussion
L 169-172: the first part is redundant, as it is very similar to the part in the introduction. I suggest rephrasing it by reasoning and connecting to it the results of your research
L 178: here and also in the introduction is missing the mention of the choros model (sensu Triantis et al. 2003), used among others by Kallimanis et al. (2011) and Fois et al. (2016)
Triantis KA, Mylonas M, Lika K, Vardinoyannis K (2003) A model for the species–area–habitat relationship. J Biogeogr 30:19–27.
And references above
L 193-196 please add at least one reference in support of this statement
L 214: Here the discussion can be deepened. For example, it can be argued that island alien species diversity is correlated with human presence (e.g. Pretto et al 2012; Dimitrakopoulos et al. 2022)
Pretto, F., Celesti-Grapow, L., Carli, E., Brundu, G., & Blasi, C. (2012). Determinants of non-native plant species richness and composition across small Mediterranean islands. Biological Invasions, 14, 2559-2572.
Dimitrakopoulos, P. G., Koukoulas, S., Michelaki, C., & Galanidis, A. (2022). Anthropogenic and environmental determinants of alien plant species spatial distribution on an island scale. Science of the Total Environment, 805, 150314
Methods
L 273: please better explain what the authors mean by 'narrow endemic taxa', are they exclusive to the island/archipelago or other distribution ranges were considered?
L 274-275: there is an error in citing the respective floras, for instance all the ones mentioned for Sicily are instead about Sardinia (for this region please consider just citing the last updates in Fois et al. 2022), while there is no one about Sicily.
Fois, M., Farris, E., Calvia, G., Campus, G., Fenu, G., Porceddu, M., & Bacchetta, G. (2022). The endemic vascular flora of Sardinia: a dynamic checklist with an overview of biogeography and conservation status. Plants, 11(5), 601.
Supplementary materials
I do not see any additional material mentioned (L 316-317), although it would be helpful to better explain at least the calculation of the topography through a text (e.g. by reporting the software, script, etc.) and figures.
References
Please revise the references and uniform them according to the guidelines. Among possible other ones, there are some discrepancies in terms of font used (e.g. L 353-355), reporting the DOIs, dots after the year (see e.g. L 348), journals sometimes in italics, sometimes not (e.g. L 342), spaces after the commas (e.g. L 348), use of n- or m- dashes (e.g. L377), use of the asterisks (e.g. L 383, 385), capital letters for journals’ names (e.g. L 371). Please also include the link to ref. 35 and consider that, generally, the date of consultation should be given for websites.
Reviewer 2 Report
Comments and Suggestions for Authors
The manuscript titled “Topographic complexity is a principal driver of plant endemism in Mediterranean Islands” submitted by Sandro Lanfranco. As reported the aim of this study was to estimate the explanatory power of a small number of variables on the species richness of vascular plants on selected Mediterranean islands and archipelagos and on the proportion of narrow endemism in each. The present study extended the notion of ‘maximum elevation’ used in other studies into that of ‘topographic diversity’ as it captures the drivers behind elevational habitat diversity more accurately than a single point measurement would. A novel approach was used where the topographic complexity and isolation of an island were estimated through more detailed methods than those utilised previously. The results demonstrated that ‘topography’, a factor that was not specifically included in previous models for Mediterranean islands, exerted a consistent, statistically significant effect on both the number of taxa as well as the proportion of endemic taxa, in all models tested. Although the factor 'isolation' was not a significant predictor of the number of taxa in any of the models, it was, however, a statistically significant predictor of the proportion of endemic taxa in two of the models.
Considering the importance of this kind of study to improve the knowledge about plant endemism in Mediterranean Islands. I believe that the manuscript is of potential interest to readers of "Plants" and falls within its scope.
Abstract: briefly summarises the major aspects of the study
Keywords: are fine.
Introduction: It is ok. It presents a well description of the state of the art introducing well the role of the proposed study in the framework of current knowledge on the subject.
Materials and Methods: Are clear and well detailed.
Results: in general, are clear.
Discussion: in general, is well-written.
Conclusion: is clear and well-written and summarize the main results observed.
Correction:
Line 141 Please GLM (Generalized linear models)
Reviewer 3 Report
Comments and Suggestions for Authors
This is a study that could be published in the Journal.
Unfortunately, the authors have made some inappropriate choices in the methods used in the analysis of data. I am highlighting some of these.
1. Topographic complexity is termed topographic diversity in several places in the text (for instance, l. 66, 178)
2. There have been earlier attempts to include some sort of complexity in Species-Area studies. See https://doi.org/10.1046/j.1365-2699.2003.00805.x.
3. I think that lumping all Balearic and Tuscan islands into one island unit is inappropriate and methodologically unsound. This misses the entire theory behind the Species – Area issue.
4. The choice of Poisson regression is inappropriate for the first series of models where the number of species is the response variable. To use Poisson regression, you must have the mean to equal variance in the observations of the response variable (spp). Here the mean μ = 187.25 and the variance σ2 = 17285.64. This causes, what is called overdispersion and invalidates the results of the regression.
Some comments in the appearance of the paper:
1. All authors are from the same department. I do not see the reason for having a different superscript (1 – 6) for each one.
2. Materials and methods should be paragraph #2. NOT #4 and after the discussion.
3. Table 1 belongs to Materials and Methods. NOT to results.
Comments on the Quality of English LanguageThis is a study that could be published in the Journal.
Unfortunately, the authors have made some inappropriate choices in the methods used in the analysis of data. I am highlighting some of these.
1. Topographic complexity is termed topographic diversity in several places in the text (for instance, l. 66, 178)
2. There have been earlier attempts to include some sort of complexity in Species-Area studies. See https://doi.org/10.1046/j.1365-2699.2003.00805.x.
3. I think that lumping all Balearic and Tuscan islands into one island unit is inappropriate and methodologically unsound. This misses the entire theory behind the Species – Area issue.
4. The choice of Poisson regression is inappropriate for the first series of models where the number of species is the response variable. To use Poisson regression, you must have the mean to equal variance in the observations of the response variable (spp). Here the mean μ = 187.25 and the variance σ2 = 17285.64. This causes, what is called overdispersion and invalidates the results of the regression.
Some comments in the appearance of the paper:
1. All authors are from the same department. I do not see the reason for having a different superscript (1 – 6) for each one.
2. Materials and methods should be paragraph #2. NOT #4 and after the discussion.
3. Table 1 belongs to Materials and Methods. NOT to results.
Round 2
Reviewer 3 Report
Comments and Suggestions for Authors
The only issue that I still have with this manuscript is the lumping of two archipelagoes. I do not find the explanations of the authors satisfactory.
Author Response
Thank you for these comments. Please see attachment.
